# Contactless Waveguide Characterization of Piezoresistive Materials for Wireless Strain Sensors

**DOI:** 10.3390/s22114085

**Published:** 2022-05-27

**Authors:** Sandra Rodini, Simone Genovesi, Giuliano Manara, Filippo Costa

**Affiliations:** Dipartimento di Ingegneria Dell’Informazione, Università di Pisa, via Caruso 16, 56122 Pisa, Italy; sandra.rodini@phd.unipi.it (S.R.); simone.genovesi@unipi.it (S.G.); giuliano.manara@unipi.it (G.M.)

**Keywords:** surface impedance, scattering parameters, strain sensor, electromagnetic bandgap

## Abstract

Stretchable materials are widely used for the realization of various sensors, but their radio frequency behavior has not been fully characterized so far. Here, an innovative method is proposed for deriving the surface impedance of this kind of materials. The material characterization represents a fundamental step for exploiting the material as a sensing element within a radio frequency device. Indeed, the proposed method is capable of retrieving the surface impedance of the material while it is being stretched, thus deriving a correspondent calibration curve. The characterization approach is based on a contactless measurement of the scattering parameters using waveguides. By exploiting the measured scattering parameters, the variation in the surface impedance as a function of both frequency and strain is recovered through an analytical inversion procedure. Numerical simulations were initially performed trough a numerical electromagnetic simulator, and subsequently, experimental validation was carried out using a dedicated test bench designed to ensure a contactless measurement of the stretchable material.

## 1. Introduction

Strain sensors represent a deeply investigated research branch in engineering, and numerous strain sensors have been designed and demonstrated [1,2,3]. The totality of these solutions require an electrical contact for sensor reading. However, in some situations (for example, in the case of critical operating conditions or in wearable devices), it would be convenient to have completely wireless strain sensors [4,5,6]. Therefore, the idea that stimulated this investigation is the development of a new class of wireless piezoresistive sensors that do not need the wiring system for reading. To do this, it is necessary to observe how the properties of the Material Under Test (MUT) change when it is stretched. Therefore, the materials used to make these sensors must be stretchable, and their properties must vary as the strain level varies [7]. The parameter that reflects the variation of the properties of the material as a function of applied strain is the surface impedance or sheet impedance [8]. Generally, the surface impedance of a material is obtained with the classic four-probe method, which allows deriving the Direct Current (DC) surface impedance [9]. This method offers good reliability, but it is a contact method and can damage the sheet. Another problem is that the characterization performed in DC may substantially deviate from the radio frequency one. Indeed, as the frequency raises, the DC approximation starts to be not rigorous, and an imaginary part is expected in the sheet impedance even if the thickness of the materials is considerably thinner than one wavelength and the hypothesis of a good conductor still holds [10]. The ideal solution to avoid damaging the material is to carry out a contactless measurement of the surface impedance. Contactless methods are also the ideal solution for estimating the surface impedance of a material that is being stretched. In the literature, there are several methods for determining the surface impedance of a material, but very few are truly a contactless method [11]. Microwave methods are mainly divided into resonant and non-resonant methods. Resonant methods [12,13,14] show good accuracy and sensitivity; however, they require an ad hoc cavity and sample preparation can be complicated. Non-resonant methods [15,16,17,18] are based on the measurement of the signal reflected and transmitted by the sample. These methods allow measurements to be made using different experimental setups and require fewer precautions [10]. In [8], a method has been proposed to derive the surface impedance of an ink deposited on a dielectric substrate by using a rectangular waveguide and an inversion procedure to derive the surface impedance from the measured scattering parameters. An improvement in the accuracy for thick supporting dielectrics has been obtained in [19]. The aforementioned methods offer a good estimate of the surface impedance of the investigated sample. However, as already pointed out, in order to fabricate a strain sensor, it is necessary to test the resistivity of the material under different stretching conditions, and thus, the piezoresistive sheet should be not in contact with the flanges of the waveguide. Here, differently from the previous methods, the waveguides are completely disconnected from the sample. Indeed, the thin sheet needs to be deformed, and, at the same time, the surface impedance must be measured. This allows obtaining the trend of the surface resistance of the material as the strain level varies and represents the calibration curve for a strain sensor. A preliminary investigation of a contactless approach was presented in [20,21,22]. In order to obtain an accurate measurement of the surface impedance, an EBG surface is positioned on one of the two flanges, which prevents field leakage in correspondence with the gap. The surface impedance value is obtained thanks to an inversion procedure starting from the scattering parameters obtained during the measurement. This method offers an accurate estimate of the real part of the surface impedance of the sheet placed between the two waveguides but gives a significant error on the imaginary part. Here, a method is proposed that allows improving the estimation and limiting the error on the imaginary part. Unlike what was carried out in [20,21], in which a calibration measurement was created without the sample, here, it is proposed to perform the calibration with the sheet, whose surface impedance measurement has been previously obtained in a closed setup. What is evaluated with the proposed method is therefore not an absolute measure of the surface impedance; the variation in the surface impedance is evaluated as a function of the applied strain, which is the needed information to derive the strain/surface impedance calibration curve.

## 2. Characterization Procedure

Starting from the preliminary analysis of the open junction discussed in [21], here, a method for the contactless characterization of stretchable materials is proposed. The proposed method consists of several steps that allow determining the variation in the surface impedance of the material when it is stretched. A sequence of three measurements was performed on the Material Under Test (MUT), and the parameters derived are summarized in Figure 1.

The first step (STEP 1) of the procedure consists of determining the surface impedance value of the stretchable materials at rest. Therefore, a closed waveguide measurement of the sample being tested is carried out, as shown in Figure 2a. The measurement in a closed waveguide allows having an accurate estimate of the surface impedance of the sheet before any stretch is applied; therefore, this value can be taken as a reference for the subsequent steps of the procedure.

The next step (STEP 2) of the procedure consists of measuring the surface impedance of the sheet in a completely contactless way at rest. To carry out the contactless measurement, two rectangular waveguides separated by an air gap are used, and the sheet is positioned in the middle so that there is no contact between the sheet and the waveguides. To reduce the field losses introduced by the air gap, an electromagnetic band-gap (EBG) surface is applied on one of the two waveguides. The measurement system is shown in Figure 2. This step is employed to characterize the parameters of the open junction represented by the two unconnected waveguides [22]. Indeed, as shown in Figure 3, the unconnected waveguides in the presence of the MUT are represented by a couple of π circuits characterized by the parameters $Y_{d1}$, $Y_{d2}$, $Z_{d}$ and $Y_{s}^{cal}$, which is the sheet impedance of the MUT calculated in the first step of the procedure.

Once the lumped parameters related to the circuit model of the junction are derived, the last step (STEP 3) of the procedure consists of measuring the sample while it is being stretched. The contactless measurement of the surface impedance of the material is carried out with different strain values in order to correlate the two quantities.

The calibration measurement carried out in STEP 2 of the proposed procedure is necessary to take into account the discontinuity between the two waveguides. Unlike what is shown in [21], the calibration measurement is performed with the sheet to be characterized in the middle. The equivalent circuit of the calibration phase is shown in the figure, in which the stretchable sheet is represented by a lumped parameter $Y_{d1}$. $Z_{d}$ and $Y_{d2}$ are the calibration parameters that take into account the discontinuity between the two waveguides. The calibration circuit can be described using the ABCD matrix:(1)$$\left(\begin{array}{cc} A & B\\ C & D \end{array}\right)=\left(\begin{array}{cc} 1 & 0\\ Y_{d1} & 1 \end{array}\right)\left(\begin{array}{cc} 1 & Z_{d}\\ 0 & 1 \end{array}\right)\left(\begin{array}{cc} 1 & 0\\ Y_{s}^{cal} & 1 \end{array}\right)\left(\begin{array}{cc} 1 & Z_{d}\\ 0 & 1 \end{array}\right)\left(\begin{array}{cc} 1 & 0\\ Y_{d2} & 1 \end{array}\right)$$
where the terms of the ABCD matrix read:(2)$$A=\frac{\left(1+S_{11}\right)\left(1-S_{22}\right)+S_{21}S_{12}}{2S_{21}}$$
(3)$$B=Z_{0}\frac{\left(1+S_{11}\right)\left(1+S_{22}\right)-S_{21}S_{12}}{2S_{21}}$$
(4)$$C=\frac{1}{Z_{0}}\frac{\left(1-S_{11}\right)\left(1-S_{22}\right)-S_{21}S_{12}}{2S_{21}}$$
(5)$$D=\frac{\left(1-S_{11}\right)\left(1+S_{22}\right)+S_{21}S_{12}}{2S_{21}}$$

Supposing we know the value of $Z_{s}$ in the unstretched state, as obtained from the closed waveguide measurement performed in STEP 1 of the procedure, and considering the relationships between the scattering parameters and the ABCD parameters, it is possible to define the analytical expression of the calibration parameters as follows:(6)$$Z_{d}=\frac{-2+\frac{\sqrt{2}\sqrt{2\;S21+\mathrm{Ys}Z0+S11\mathrm{Ys}\mathrm{ZO}-S21^{2}\mathrm{Ys}\mathrm{ZO}+S22\mathrm{Ys}\mathrm{ZO}+S11\;S22\mathrm{Ys}\mathrm{ZO}}}{\sqrt{S21}}}{2\mathrm{Ys}}$$
(7)$$Y_{d1}=\frac{1-S11-2S21+S21^{2}+S22-S11S22-2S21YSZ}{2S21Z(2+YsZ)}$$
(8)$$Y_{d2}=\frac{1+S11-2\;S_{21}+S_{21}^{2}-S_{22}-S_{11}\;S_{22}-2\;S_{21}Y_{s}Z}{2\;S_{21}Z\left(2+Y_{s}Z\right)}$$

In the third step of the procedure, the sheet of material is stretched to various levels of strain. The objective is to obtain the new surface impedance value of the material in the stretched state. Now, the equivalent circuit is shown in Figure 3b. The difference with respect to the unstretched circuit mode is that in the second circuit the surface impedance of the MUT is different from the previous one. Considering all the other parameters unchanged with respect to the unstretched measurement performed in STEP 2, it is possible to obtain the new surface impedance value through an inversion procedure starting from the new $ABCD^{\prime }$ matrix:(9)$$\left(\begin{array}{cc} A^{{}^{\prime }} & B^{{}^{\prime }}\\ C^{{}^{\prime }} & D^{{}^{\prime }} \end{array}\right)=\left(\begin{array}{cc} 1 & 0\\ Y_{d1} & 1 \end{array}\right)\left(\begin{array}{cc} 1 & Z_{d}\\ 0 & 1 \end{array}\right)\left(\begin{array}{cc} 1 & 0\\ Ys & 1 \end{array}\right)\left(\begin{array}{cc} 1 & Z_{d}\\ 0 & 1 \end{array}\right)\left(\begin{array}{cc} 1 & 0\\ Y_{d2} & 1 \end{array}\right)$$
(10)$$Z_{s}=\frac{S_{21}^{\prime }\left(Z+Z_{0}+Y_{1}ZZ_{0}\right)\left(Z+Z_{0}+Y_{2}ZZ_{0}\right)}{-2S_{21}^{\prime }Z+2Z_{0}-2S_{21}^{\prime }Z_{0}-2S_{21}^{\prime }Y_{1}ZZ_{0}-2S_{21}^{\prime }Y_{2}ZZ_{0}-2S_{21}^{\prime }Y_{1}Z_{0}^{2}-2S_{21}^{\prime }Y_{2}Z_{0}^{2}-2S_{21}^{\prime }Y_{1}Y_{2}ZZ_{0}^{2}}$$
where $Z_{s}$ is the surface impedance value when the material is stretched by ϵ = ΔL/$L_{0}$ ($L_{0}$ is the initial dimension of the sample, and ΔL is equal to $\left(L-L_{0}\right)$.

In order to verify the assumption that the calibration parameters remain unchanged with respect to the unstretched measurement, the field distribution in the discontinuity is analyzed for two different values of the MUT sheet impedance. By observing Figure 4, it can be observed that even if the value of the surface impedance of the sheet placed between the two waveguides varies, the field distribution does not have considerable variations. This justify the assumption that $Z_{d}$, $Y_{d1}$ and $Y_{d2}$ remain constant before and during the elongation of the material.

## 3. Results

In order to understand the benefits of the proposed calibration technique of the junction obtained by putting the sheet between the two unconnected waveguides, some electromagnetic numerical simulations were performed using the CST Studio Suite Software. Subsequently, the characterization technique was tested on a real test bench to confirm its reliability.

### 3.1. Numerical Results

Two Anritsu^®^ rectangular waveguides (WR137) with internal dimensions of 34.85 mm × 15.80 mm were used in the simulations. These waveguides work optimally in the frequency range between 5.85 GHz and 8.20 GHz. To validate the model, a first simulation was carried out using a sheet with a known surface impedance of 100 Ω/sq. The results were obtained from the simulations with CST Studio Suite Software (Simulia Company). By elaborating the scattering parameters obtained from the simulation according to Equations (Equation 6)–(Equation 8), we obtained the calibration parameters $Z_{d}$, $Y_{d1}$ and $Y_{d2}$. After this first simulation, a parametric variation in the sheet impedance (from 50 Ω/sq to 120 Ω/sq) was carried out. Using the $S_{21}$ parameter of these simulations and the calibration parameters obtained in the previous step, the new sheet impedance was extracted using Equation (Equation 10).

Figure 5 shows the results of the numerical simulations performed with the CST Studio Suite Software, in which a comparison is made between the real value with which the simulations were performed and the value obtained using the proposed inversion procedure. In the figure, Expected means the surface impedance value that we have entered into the software to carry out the numerical simulations. On the other hand, Estimated represents the surface impedance value obtained by inserting the scattering parameters obtained from the simulations into Formula (Equation 10). From Figure 5, we can see that when using a surface impedance sheet equal to 100 Ω/sq for the calibration, it is possible to accurately estimate the surface impedance of unknown sheets. In this case, the estimate of the surface impedance of three sheets with a surface impedance between 50 Ω/sq and 120 Ω/sq is shown. The estimate of the real part is very accurate as well as the imaginary part. This is an improvement with respect to the previous calibration approach without the sample (empty calibration). The fact that a calibration sheet with impedance equal to 100 Ω/sq can be used to accurately estimate the $Z_{s}$ of sheets with surface impedance between 50 Ω/sq and 120 Ω/sq suggests that this method is optimal for estimating the variation surface impedance of a sheet as its strain level varies.

### 3.2. Experimental Results

The proposed method was validated through measurements carried out on a real test bench. The experimental setup was developed in such a way as to guarantee a real contactless measurement of the surface impedance of the sample.To carry out the measurements, an Anritsu^®^ Shockline MS46524B Vector Network Analyzer (VNA) working up to 43 GHz and two WR137 rectangular waveguides operating in the frequency range 5.85–8.20 GHz were used. An EBG surface was positioned on one of the waveguides. This surface was made using a traditional photolithographic method on a FR4 substrate with a thickness equal to 1.6 mm. The supports for the waveguides were made of Stratasys ABS-M30 (acrylonitrile butadiene styrene) material using 3D printing techniques (Stratasys 3D printer).The custom design of these pieces guarantees a correct gap between the two waveguides (7 mm) to ensure that the material being tested does not come into contact with them. In order to be able to stretch the material by a precise amount, an Arduino^®^ UNO controlled IGUS drylin^®^ stainless steel TR10x2 lead screw was used. The whole measuring system is automated. Arduino is connected to Matlab^®^ software (MathWorks Company) so that it is possible to give the input for the elongation of the sample, and once the sample has been elongated, again through Matlab, the estimate of the surface impedance is performed using the scattering parameters acquired using the VNA (also connected to the computer and controlled via Matlab). The measurement system is shown in Figure 6a. By using the Arduino controlled rail, it is possible to control the strain of the sample in millimeters. Moreover, the measurement is highly repetitive, and more measurement cycles can also be performed. The material chosen to test the proposed measurement system is Shieldex^®^ Med-tex P130 [23], a silver plated knitted fabric shows in Figure 6b.

As discussed in Section 2, the first step was to measure the surface impedance of the material in a closed waveguide. This measure, being reliable, was taken as a reference for the subsequent steps. Subsequently, the waveguides and the material were placed in the appropriate ABS supports, and the surface impedance at rest was initially measured to estimate the parameters of the π circuits representing the junction. After this second step, the material at different elongation levels was measured. We have chosen different values of a displacements in order to evaluate the change in surface impedance of the material. After each step, the surface impedance of the material is extracted according to the inversion procedure discussed in Section 2. Figure 7a,b show the surface impedance for different elongation values. In Figure 7c,d, the results of the numerical simulations performed with the CST software are shown. In order to compare the measured results with the simulated one for this specific sample, simulations with surface impedance values varying in the range between 20 and i15 Ohms/sq and 40-i15 Ohms/sq were carried out. The range of values employed in the simulation is set according to the estimations obtained for the sample here characterized. When comparing Figure 7a,b with Figure 7c,d, it is possible to observe that the obtained results are characterized by the same behaviour.The difference between experimental results and simulations are due to the experimental setup. In fact, in the simulation, an ideal situation is represented in which the material remains taut, and undulations do not form on it; therefore, the material always remains equidistant between the two waveguides. In reality, the material is is an elastic fabric; therefore, as it is stretched, undulations are created on it, and the distance between the material and the guides does not remain exactly the same. These variations during the measurement are reflected in an imperfect correspondence between experimental results and simulation results. Compared to the previous calibration approach based on empty measurement calibration [21], the error on the imaginary part is considerably smaller. If we consider a surface impedance value equal to 20 Ω/sq, the error on the imaginary part was greater than 30%. With the method proposed in this work for a surface impedance value of 20 Ω/sq, the error on the imaginary part is less than 10%.

Finally, the obtained results allow us to build the calibration curve to design a strain sensor. In order to obtain the calibration curve, we choose a frequency within the operating range of the waveguide used. For instance, by selecting the frequency of 6.2 GHz, the maximum variation of the real part of the surface impedance is observed. We can then set the frequency at 6.2 GHz and extract the surface impedance values at this frequency value and for different strain values. The calibration curve for the material being tested is shown in Figure 8.

## 4. Conclusions

A novel approach for estimating the surface impedance of piezoresistive materials has been presented. The method is based on a contactless measurement of the sample with two unconnected waveguides. With the proposed setup, the scattering parameters are measured while the sample is stretched. The accuracy of the proposed approach has been initially evaluated using simulated results and subsequently by measuring real piezoresistive samples. The method relies on an initial calibration procedure based on the measurement of a thin sheet with a known surface impedance. The initial value is estimated with a closed waveguide setup. In the second phase, surface impedance measurements were made while the sheet is stretched. Differently from an empty calibration approach, the proposed extraction procedure allows obtaining a good estimate of both the real part and the imaginary part of the surface impedance. This characterization of the sample in terms of sheet impedance variation represents a fundamental element for the realization of the calibration curve for designing a radio frequency strain sensor. The calibration curve relates the elongation of the material to the variation of the surface impedance of the material being tested.

## Figures and Tables

**Figure 1 sensors-22-04085-f001:**
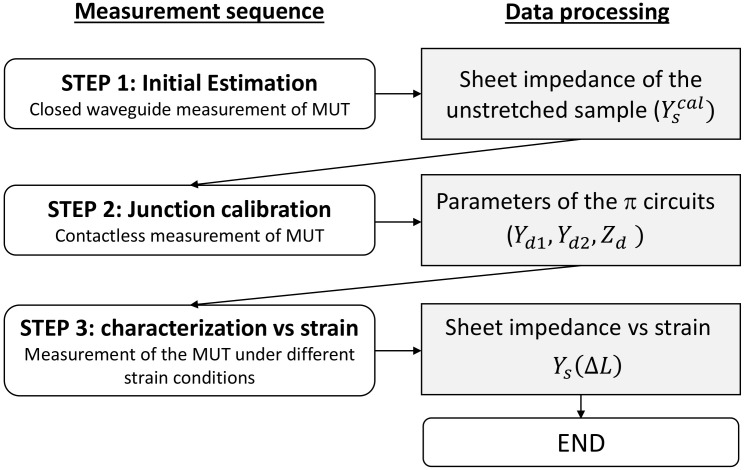
Block diagram of the proposed approach for the characterization of piezoresistive materials. MUT: Material Under Test.

**Figure 2 sensors-22-04085-f002:**
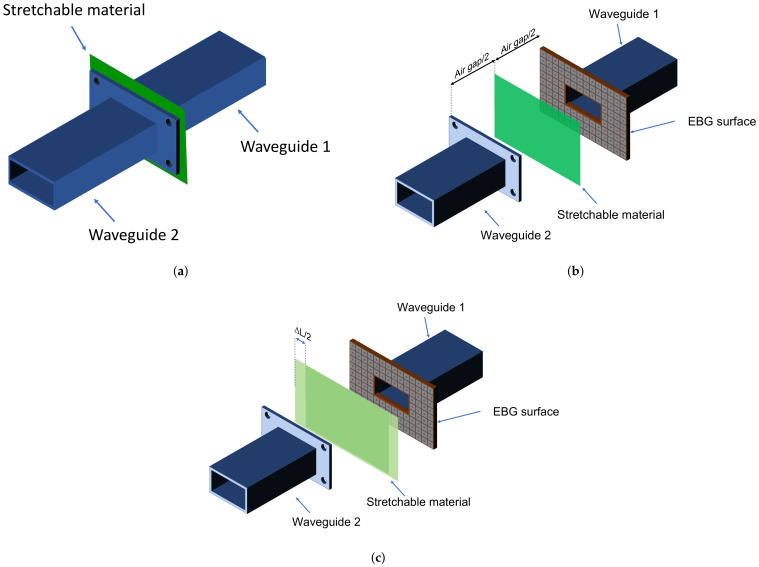
(**a**) STEP 1: Closed waveguide measurement; (**b**) STEP 2: Calibration setup with the material in the unstretched phase in the middle; (**c**) STEP 3: Measurement setup for the contactless estimation of the surface impedance when the material has been stretched by delta L.

**Figure 3 sensors-22-04085-f003:**
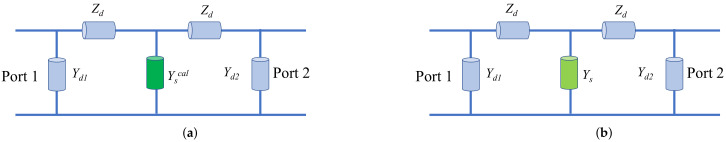
Equivalent circuit model with material in the unstretched state (calibration circuit) (**a**) and with material in stretched state (**b**).

**Figure 4 sensors-22-04085-f004:**
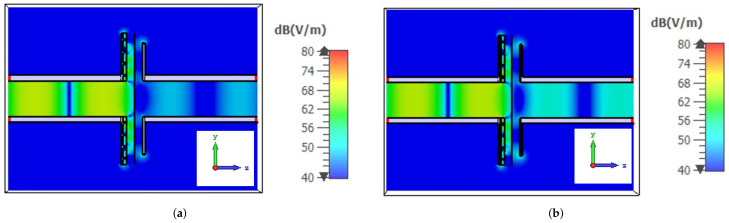
Field distribution at the discontinuity: (**a**) field distribution with the calibration sheet (surface impedance value equal to 50 Ω/sq); (**b**) field distribution using a sheet with a surface impedance equal to 100 Ω/sq.

**Figure 5 sensors-22-04085-f005:**
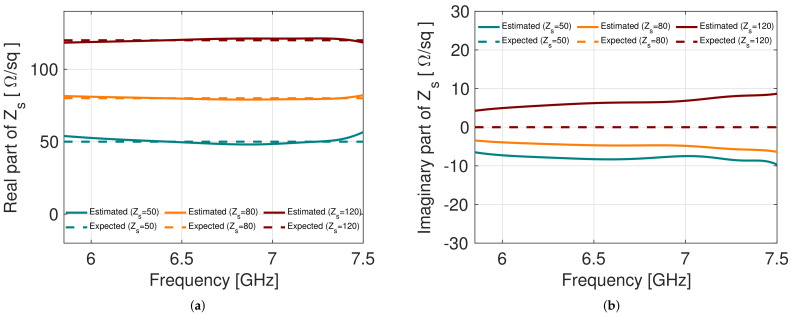
(**a**) Real part of the surface impedance; (**b**) imaginary part of the surface impedance when a 100 Ω/sq impedance sheet has been used for calibration.

**Figure 6 sensors-22-04085-f006:**
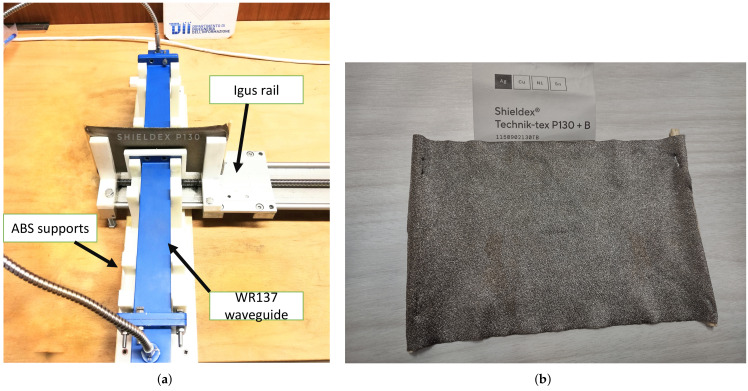
(**a**) Experimental setup; (**b**) stretchable material.

**Figure 7 sensors-22-04085-f007:**
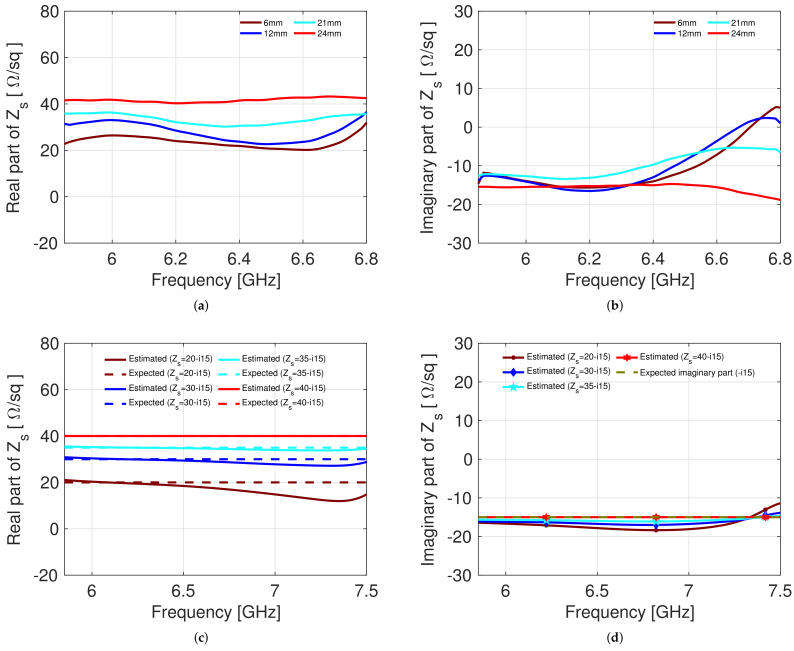
Comparison between experimental measurements and numerical simulations: (**a**) real part of the surface impedance obtained from the experimental measurements; (**b**) imaginary part of the surface impedance obtained from the experimental measurements; (**c**) real part of the surface impedance obtained from the CST simulations; (**d**) imaginary part of the surface impedance obtained from the CST simulations.

**Figure 8 sensors-22-04085-f008:**
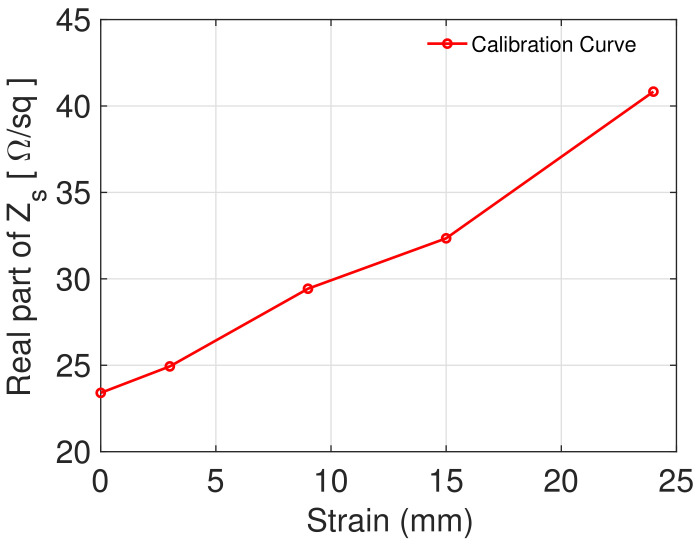
Calibration curve for the material under test.

## Data Availability

Not appliable.

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
