# Peer review of "Contactless Waveguide Characterization of Piezoresistive Materials for Wireless Strain Sensors"

_sensors, 2022, doi:10.3390/s22114085_

Round 1

Reviewer 1 Report

The manuscript entitled “Contactless Waveguide characterization of Piezoresistive Materials for Wireless Strain Sensors” under review present an innovative method is proposed that allows deriving the surface impedance of stretchable materials. I recommend to address the following minor revisions, which are required before publishing.

  1. The authors could mention more clearly the novelty and purpose of the work proposed in the introduction section and it is now known that they will address the readers of Sensors journal. One can add, suggest some specific applications in the real environment (not just demonstrations in the laboratory) of the proposed measurement technique.
  2. The authors presented a limited literature, I consider this due to an undefined intention at this stage to address the readers of Sensors journal. Forty percent of the reference works are produced by the same research group that produced this work. This does not mean that this work does not bring a real and original contribution in the field. However, I believe that a balance will contribute to the value of this work, giving readers a broader picture of possible applications.
  3. The author must define the acronyms in the first place when they appear.

Row 87 – EBG surface

  1. Specify the source for the materials and software packages (Company, ...). This information is very important for readers who want to replicate this very interesting experimental setup.

Row 134 - Two rectangular waveguides (WR137)

Row 138 - CST Studio Suite Software.

Row 166 - IGUS rail was used

Row 167 – Matlab

Row 176 - Shieldex Med-tex P130

Row 179 - ABS supports

In conclusion, a minor revision would increase the value of communication.

Author Response

Comment R1.1 The authors could mention more clearly the novelty and purpose of the work
proposed in the introduction section and it is now known that they will address the readers of
Sensors journal. One can add, suggest some specific applications in the real environment (not just
demonstrations in the laboratory) of the proposed measurement technique.
Answer to R1.1 The manuscript describes a laboratory measurement technique to be used in
the laboratory to pre-characterize materials and obtain a strain-sheet impedance calibration curve.
These materials can subsequently be used to make strain sensors. we underlined it in the new version
of the manuscript. Based on the calibration curve obtained trough the proposed measurement
method, a wireless Strain sensor may be then designed for practical applications. Currently in the
literature there are no techniques for the radio-frequency characterization of materials without a
contact with the material under test. As indicated in the introduction, most of the techniques
involve the use of electrodes for the measurement of surface impedance. Here we propose an improvement
of a proposed measurement technique in which we try to improve the estimation of the
surface impedance, in particular of the imaginary part.

Comment R1.2 The authors presented a limited literature, I consider this due to an undefined
intention at this stage to address the readers of Sensors journal. Forty percent of the reference
works are produced by the same research group that produced this work. This does not mean that
this work does not bring a real and original contribution in the field. However, I believe that
a balance will contribute to the value of this work, giving readers a broader picture of possible
applications.
Answer to R1.2 Thanks for the tip. We have expanded the literature by including the following
references:
[12]Hernandez, A.; Martin, E.; Margineda, J.; Zamarro, J. Resonant cavities for measuring the
surface resistance of metals at X-band frequencies. Journal of Physics E: Scientific Instruments
1986, 19, 222.
[13]Krupka, J.; Klinger, M.; Kuhn, M.; Baryanyak, A.; Stiller, M.; Hinken, J.; Modelski, J.
Surface resistance measurements of HTS films by means of sapphire dielectric resonators. IEEE
Transactions on Applied Superconductivity 1993, 3, 3043–3048.
[14]Krupka, J.; Strupinski, W. Measurements of the sheet resistance and conductivity of thin
epitaxial graphene and SiC films. Applied Physics Letters 2010, 96, 082101.
[15]Booth, J.; Wu, D.H.; Anlage, S.M. A broadband method for the measurement of the surface
impedance of thin films at microwave frequencies. Review of scientific instruments 1994, 65,
2082–2090.
[16]Ye, M.; Tariq, R.U.; Zhao, X.L.; Li, W.D.; He, Y.N. Contactless Measurement of Sheet
Resistance of Nanomaterial Using Waveguide Reflection Method. Materials 2020, 13, 5240.
[17]Feng, Y.R.; Wei, X.C.; Yi, D.; Gao, R.X.K. An enhanced one-port waveguide method
for sheet resistance extraction. IEEE Transactions on Electromagnetic Compatibility 2019, 62,
1822–1829.
[18]Lee, M.J.; Collier, R. The sheet resistance of thin metallic films and stripes at both DC
and 130 GHz. Microelectronic engineering 2004, 73, 916–919.

Comment R1.3 The author must define the acronyms in the first place when they appear. Row
87 – EBG surface Electromagnetic Band-Gap

Answer to R1.3 Thanks for the suggestion. We have now inserted the acronym.
Comment R1.4 Specify the source for the materials and software packages (Company, ...). This
information is very important for readers who want to replicate this very interesting experimental
setup. Row 134 - Two rectangular waveguides (WR137); Row 138 - CST Studio Suite Software;
Row 166 - IGUS rail was used; Row 167 – Matlab; Row 176 - Shieldex Med-tex P130; Row 179 -
ABS supports.
Answer to R1.4 We inserted the source for materials and software.
-Row 134 - Two rectangular waveguides (WR137) has been replaced with ”Two Anritsu®
rectangular waveguides (WR137) with internal dimensions 34.85 mm x 15.80 mm were used in the
simulations”.
-Row 138 - CST Studio Suite Software;
The results were obtained from the simulations with CST Studio Suite Software (Simulia Company)
-Row 166 - IGUS rail was used;
In order to be able to stretch the material by a precise amount, an Arduino® UNO controlled
IGUS drylin® stainless steel TR10x2 lead screw was used.
-Row 167 – Matlab;
Arduino is connected to Matlab® software (MathWorks Company) so that it is possible to
give the input for the elongation of the sample and once the sample has been elongated, again
through Matlab, the estimate of the surface impedance is performed using the scattering parameters
acquired using the VNA (also it connected to the computer and controlled via Matlab).
-Row 176 - Shieldex Med-tex P130;
The website for the material has been added [23]Shieldex®Med-tex P130. https://www.shieldex.de/products/tex-p130/. Accessed: 2022-05-16.
-Row 179 - ABS supports.
The supports for the waveguides were made of Stratasys ABS-M30 (acrylonitrile butadiene
styrene) material using 3D printing techniques (Stratasys 3D printer).
3

Reviewer 2 Report

The authors demonstrate interesting computer simulation and measurements of piezoresistive materials for wireless strain sensors.  The manuscript is well written in English language. Some confusing illustrations need to be clarified prior to an accept for publication.

  1. in Figure 5 and Figure 7, what is the difference between Estimated value and Expected value?  It seems there is no clear explanation in the main texts.
  2. Figure 7 (b) and Figure 7(d), considerable difference can be observed between experiment and simulation,  why not give more explanations about the reason behind?

Author Response

Comment R2.1 In Figure 5 and Figure 7, what is the difference between Estimated value and
Expected value? It seems there is no clear explanation in the main texts.
Answer to R2.1 Fig. 5 and Fig. 7 (c) and 7 (d) represent the results of numerical simulations
performed with the CST Studio Suite Software. By ’Expected value’ we mean the surface
impedance value that we have entered into the software to carry out the numerical simulations. On
the contrary, with ’Estimated value’ we mean the surface impedance value obtained by inserting
the scattering parameters obtained from the simulations in the formula [10]. Therefore in these
figures we want to make a comparison between the real value with which the simulations were
performed and the value obtained using the proposed inversion procedure.

Comment R2.2 Figure 7 (b) and Figure 7(d), considerable difference can be observed between
experiment and simulation, why not give more explanations about the reason behind?
Answer to R2.2 As indicated in the text, the difference between experimental results and simulations
are due to the fact that in the simulation an ideal situation is represented in which the
material remains taut and without undulations, therefore the material always remains equidistant
between the two waveguides. During experiments, being the material flexible, waves can occur on
it as it is stretched,and the distance between the material and the waveguides does not remain
exactly the desired one. These variations during the measurement are reflected in an imperfect
correspondence between experimental results and simulation results. However, it is important to
note that the starting value at 5.85 GHz is in both cases around -i15.

Round 2

Reviewer 2 Report

The authors have made careful corrections following my review comments, I thereby suggest an Accept of the revised manuscript for publication in the journal of Sensors.